# Scalable hybrid quantum Monte Carlo simulation of U(1) gauge field coupled to fermions on GPU

Kexin Feng,[1, *] Chuang Chen,[1] and Zi Yang Meng[1, †]

[1]*Department of Physics and HK Institute of Quantum Science & Technology,*
*The University of Hong Kong, Pokfulam Road, Hong Kong, China*
(Dated: November 7, 2025)

The problem of a $U(1)$ gauge field coupled to fermions in (2+1) dimensions is of fundamental importance, as it is believed to give rise to $U(1)$ Dirac spin liquid (DSL) state in quantum magnets and the strongly coupled conformal field theory in quantum electrodynamics. However, numerical progress has long been hampered by the steep computational cost of traditional determinant quantum Monte Carlo (QMC). Here, we develop a GPU-accelerated hybrid QMC algorithm assisted with several novel technical improvements, which significantly reduces the complexity to be linear with respect to space-time volume, and scales the simulation up to an unprecedented size. At this scale, we observe asymptotic convergence of fermion bilinear correlator and conserved current correlator, which was unclear previously, and find their scaling dimensions in good agreement with field theory, which supports the conformal nature of the Dirac spin liquid. Our advances establish a scalable framework to study DSL physics and its transitions at larger scales.

## INTRODUCTION

The emergence of Dirac spin liquid (DSL), i.e. an emergent $U(1)$ gauge field coupled to gapless relativistic spinons in (2+1)d [1–11], or, in high-energy physics terms, a deconfined phase in 3D quantum electrodynamics (QED$_3$) with $N_f$ flavors of massless fermions [12–19], has attracted continuous attention from condensed matter and high-energy physicists over the past decades. For sufficiently large $N_f$, QED$_3$ is expected to flow to a conformal field theory (CFT) in the infrared [16, 18], containing a variety of scaling operators including fermion bilinear currents, mass operators, and monopole operators. Such a critical state has been argued to emerge in frustrated spin systems, where no magnetic order appears and the spinons acquire Dirac dispersions, and their interactions are mediated by fluctuating $U(1)$ gauge fields [1–11, 20–34]. In this condensed matter context, the gapless and critical ground state is referred to as the $U(1)$ DSL.

From the field-theoretical viewpoint, the basic structure of QED$_3$ CFT is understood—one knows the set of primary operators and rough estimates of their scaling dimensions [16, 17, 19]. Such quantitative information about the scaling dimensions of fermion bilinear operators is important, since the scaling dimensions govern the asymptotic decay of correlation functions and diagnose the conformal nature of the DSL. In particular, the spin, bond, and dimer correlations correspond to different fermion bilinears that transform in the adjoint representation of the emergent $SU(4)$ symmetry for $N_f = 2$, and are expected to share the same scaling exponent [3, 35]. Establishing this universality and comparing with analytical approaches such as large-$N_f$ expansion or $\epsilon$-expansion [16, 17] is crucial to confirm the conformal fixed point.

On the condensed matter experimental side, signatures of proximate DSLs have been reported in frustrated magnets, including triangular lattice antiferromagnets YbZn$_2$GaO$_5$ [36, 37] and A-YbSe$_2$ delafossites [38, 39], as well as kagomé antiferromagnets [40–46]. For example, the recently synthesized kagomé compound YCu$_3$(OD)$_6$Br$_2$[Br$_{1-x}$(OD)$_x$] shows

a clear $T^2$ contribution in specific heat [40], inelastic neutron scattering reveals a continuum consistent with Dirac cones [41, 42], and magnetic torque detects unconventional oscillations near the 1/9 plateau [46]. These results point toward DSL behavior, but a quantitative comparison requires reliable knowledge of the scaling of correlation functions.

On the computational side, progress has been hindered by the absence of efficient and unbiased methods. Determinant quantum Monte Carlo (DQMC) has been applied to finite-size lattice QED$_3$ [6, 7, 11], but its computational cost scales as $O(N_\tau V_s^3)$, with $N_\tau$ the imaginary-time extent and $V_s$ the spatial volume. This cubic scaling, together with slow dynamics of local gauge updates, restricts simulations to small lattices and low precision. In critical phases such as the DSL, DQMC also suffers from severe autocorrelation effects [47], further reducing efficiency. As a result, DQMC has not been able to access sufficiently large scales to settle questions about operator scaling dimensions or to confirm the conformal properties of QED$_3$.

Hybrid quantum Monte Carlo (HQMC) offers an appealing alternative [48–51]. Unlike local-update DQMC, HQMC updates gauge field configurations globally using Hamiltonian dynamics, while the pseudofermion representation replaces explicit determinant evaluations by iterative solutions of linear systems. In principle this reduces the complexity from $O(N_\tau V_s^3)$ to $O(N_\tau V_s)$, and is inherently well suited to parallelization. Yet in earlier condensed matter applications, HQMC faced difficulties such as poor matrix conditioning and diverging iteration counts [48, 52], which limited its practical advantages. More recently, HQMC has been successfully applied to spin-fermion models [53, 54], where optimizations including preconditioned conjugate gradient solvers, matrix-free multiplications, and GPU acceleration enabled large-scale simulations. With these computational improvements, the non-Fermi-liquid scaling in the fermionic self-energy and bosonic propagators (space-time correlation functions) are observed. These encouraging developments motivate a dedicated HQMC approach to lattice QED$_3$.

Inspired by this progress, in this paper we apply HQMC

to simulate a $U(1)$ gauge field model coupled to $N_f = 2$ fermions in (2+1)d, fully accelerated on GPUs. We scale the simulation size up to $N_\tau \times L \times L = 660 \times 66 \times 66$, which represents a major improvement over previous DQMC studies of this model [6, 7, 11] (typically limited to $200 \times 20 \times 20$), and exceeds by more than a factor of three the largest HQMC applications in spin-fermion models [53, 54]. This advance is achieved through three problem-specific GPU optimizations:

- a tailored preconditioner that stabilizes and accelerates the conjugate gradient solver,

- customized CUDA kernels for matrix–vector multiplication exploiting the sparsity pattern of the fermion matrix, and

- CUDA graph compilation to minimize kernel-launch overhead.

With these optimizations, the computational complexity is reduced from DQMC's $O(N_\tau V_s^3)$ scaling to nearly linear $O(N_\tau V_s)$, allowing us to reach an unprecedented regime of system sizes. At this scale, we are able to investigate the correlation functions of various fermion bilinear operators and conserved currents with asymptotic accuracy. We find that spin and bond correlations share the same scaling dimension in the thermodynamic limit, consistent with their identification as adjoint bilinears of the emergent $SU(4)$ symmetry. The extracted scaling dimension $\Delta_{\text{adj}}$ lies in the range 1.75–1.95, in agreement with the most recent analytical $\epsilon$-expansion results [17]. We further observe that flux–flux correlations are governed by the conserved current dimension $\Delta_J = 2$, leading to asymptotic long-time decay $\sim 1/\tau^4$. These results provide new, unbiased evidence for the conformal nature of the $U(1)$ DSL in QED$_3$. Beyond these physical findings, our computational developments establish a scalable framework to explore DSL stability and its transitions to symmetry-breaking phases with significantly reduced computational cost.

<b>RESULTS</b>

**Model**

The models we study are quantum electrodynamics (QED$_3$) defined on a cubic space-time lattice (square spatial lattice), coupled with two-flavor fermions, as shown in Fig. 1 [6, 7]. Here, we start with the action of these lattice QED$_3$ models: $S = S_F + S_B = \int_0^\beta d\tau \, (L_F + L_B)$, where $\beta$ is the inverse temperature and $\tau$ is the imaginary time. The first term

$$L_F = \sum_{\langle ij \rangle, \alpha} \psi_{i,\tau,\alpha}^\dagger \left( \partial_\tau \delta_{ij} - t e^{i\phi_{ij,\tau}} \right) \psi_{j,\tau,\alpha} + \text{h.c.} \quad (1)$$

is the fermionic Lagrangian, where $\psi_{i,\tau,\alpha}$ is the fermionic field with spatial index $i \in [1, L]$, temporal index $\tau \in [1, N_\tau]$ and flavor index $\alpha \in \{\uparrow, \downarrow\}$. $t = 1$ is the hopping amplitude and set as the energy unit throughout the paper. $\phi_{ij,\tau}$ is a bosonic scalar that lives on a bond $ij$, and acts as a $U(1)$ gauge

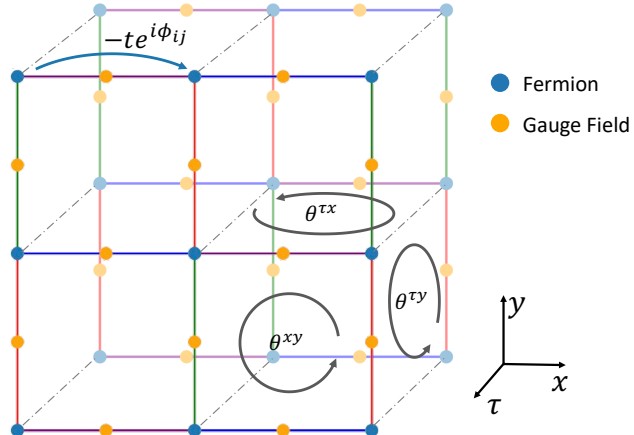

FIG. 1. **Lattice model of QED$_3$.** Fermions live on the sites of the lattice represented by blue dots, while gauge fields live on the bonds of the lattice represented by yellow dots. Fermion hops between nearest-neighbor sites with a phase $e^{i\phi_{ij}}$. The plot shows two adjacent square lattice layers, which are connected with temporal bonds represented as black dashed line. Gauge fields on these temporal bonds are set to be zero. Within a square lattice layer, the four bond colors represent the four families of bonds, which will be used in the checkerboard decomposition, as shown in Eq. (22); within each family, the fermion hopping terms represented by the bonds all mutually commute. In the bottom right cube, the circles on the facets denote the positive directions of the magnetic flux going out of the cube. The magnetic flux through a space-time plaquette $\square$ is defined as $\theta^{\mu\nu} = \sum_{ij \in \text{edge}(\square)} \phi_{ij}$, where $\mu, \nu \in \{x, y, \tau\}$ denote the plaquette orientation, $ij$ are along the positive direction, and $\phi_{ij} = -\phi_{ji}$.

field. The dynamics of the $U(1)$ gauge field is described by the bosonic action $S_B$.

In this paper, we focus on a dubbed non-compact non-compact $U(1)$ gauge field described by the Lagrangian

$$L_B^{\text{nc}} = \frac{1}{J\Delta\tau^2} \sum_{\langle ij \rangle} \left[ \phi_{ij,\tau+1} - \phi_{ij,\tau} \right]^2 + K \sum_\square \cos(\text{curl}\,\phi), \quad (2)$$

where $J$ and $K$ are the coupling constants, $\Delta\tau = 0.1$ is the imaginary time step introduced in the Trotter decomposition [6], and $\text{curl}\phi = \sum_{ij \in \text{edge}(\square)} \phi_{ij}$ is a discretized curvature of the gauge field on a spacial plaquette as shown in Fig. 1. The name non-compact comes from the fact that the gauge flux, defined as the discretized curvature of the gauge field [12, 55] on a face of the space-time cube illustrated in Fig. 1, does not have $2\pi$-periodicity, as opposed to the compact Lagrangian, e.g. $L_B^c = \frac{1}{J\Delta\tau^2} \sum_{\langle ij \rangle} \left[ 1 - \cos \left( \phi_{ij,\tau+1} - \phi_{ij,\tau} \right) \right] + K \sum_\square \cos(\text{curl}\,\phi)$, where the gauge flux does have $2\pi$-periodicity (see methods section). As a result, the non-compact model does not contain gauge monopoles, which gives a clean background to study the deconfined U(1) Dirac spin liquid phase in this model [6].

After tracing out the quadratic fermion field $\psi$ in Eq. (1),

we obtain the partition function

$$Z = \int [\delta\phi] \, e^{-S_B(\phi)} \prod_\alpha \det M_\alpha(\phi), \qquad (3)$$

where $M_\alpha$ is large sparse matrix of size $N_\tau V_s \times N_\tau V_s$, where $V_s = L \times L$ is the number of spatial lattice sizes and $N_\tau$ is the number of imaginary time slices. In our simulation, $N_\tau = 10L$ ($\beta = \frac{1}{T} = L$ and $\Delta\tau = 0.1$) to ensure that the temperature is low enough for a given $L$. Here, the $\det M_\alpha(\phi)$ has cubic time complexity and is the major bottleneck in scaling up the simulation. The implementation of the partition function in space-time path-integral in QMC is given in Methods section.

**GPU acceleration and the problem-specific preconditioner**

To efficiently sample the gauge field described by the partition function Eq. (3) at large scale, we apply HQMC algorithm aided by several optimization techniques, including the specially designed preconditioner tailored to this model, the customized CUDA kernel with local caching, and the CUDA Graph compilation technique. These three techniques are crucial in accelerating our simulation, and have broad potential impact in high-performance computation.

First, our customized CUDA kernel is designed to effectively exploit the sparsity pattern of the large matrices, and puts the reusable matrix entries in the shared memory of GPU, which is also called local caching [54]. In this way, the communication overhead within GPU is significantly reduced, and a 2.9 - 3.8 times speedup is achieved compared to naive GPU implementation, which is shown in Fig.2(a). This local caching is an advanced optimization technique in large language model system [56], and now finds its application in broader scientific computing community [53, 54].

Second, CUDA graph is another important optimization technique in our simulation. This technique was not introduced until very recently, has been applied in a variety of large language model systems [57, 58], and just starts to find its application in the broader scientific computing field. This technique compiles a series of CUDA kernel launches into a single CUDA graph, which only needs to be launched once. Thus, the large amount of repetitive CUDA kernel launching overhead is spared (see methods section for details). As illustrated in Fig.2(b), this CUDA graph optimization achieves a speed up of around 2-5 times on moderate lattice sizes (around $L < 30$) and 1-2 times on large system sizes, compared to the baseline where CUDA Graph is turned off. The technical details are presented in Methods section.

Third, finding an efficient preconditioner is challenging. There is no universal way to compute it, and is still an ongoing research [59, 60] on its own. In addition, when scaling up the simulation to $N_\tau \times L^2 = 660 \times 66^2$, we have to face extra challenge of memory bottleneck when computing the preconditioner, which could push the peak memory up to thousands of gigabytes. However, specific to the problem studied here, we find a proper preconditioner that successfully addresses all the above challenges. The key relies on the two-fold generalizability in our preconditioner. The first is that the preconditioner

obtained by one gauge configuration can be globally applied on other generic gauge configurations, without the need of local recomputation. The second is that the entries of the preconditioner at large lattice sizes can be directly inferred from the small lattice sizes, without any computation (see Methods section). These findings can be applied to the study of other problems involving the gauge-field-connected fermion hopping Lagrangian like Eq. (1).

Putting these together, we successfully lower the time complexity of the HQMC simulation down to a linear complexity $O(V_s N_\tau)$, which is shown in Fig. 2(c). The simulation size of this problem is pushed up to an unprecedented scale of $N_\tau \times L \times L = 660 \times 66 \times 66$. In Fig. 2(c), we also show the latency of the DQMC, which has a time complexity of $O(L^7)$ represented by the green dashed line. At system size of $L^3 \sim 10^5$, the latency of HQMC is two orders of magnitude smaller than that of DQMC, which shows clear advantage of the HQMC.

**Physical observables and operator scaling dimension**

We now introduce the fermionic observables to measure in the HQMC sampling, which will be used to characterize and detect the $U(1)$ DSL phase, which includes the following two gauge-invariant fermion bilinears correlation functions – spin-spin and bond-bond correlation functions:

$$C_S^{\alpha\beta}(i, j) = \langle S^{\alpha\beta}(i) S^{\beta\alpha}(j) \rangle \qquad (4)$$

$$C_B(i, j) = \langle B_i B_j \rangle - \langle B_i \rangle \langle B_j \rangle. \qquad (5)$$

Here, the spin operator is defined as $S^{\alpha\beta}(i) = \psi_{i\alpha}^\dagger \psi_{i\beta} - \frac{1}{N_f} \delta_{\alpha\beta} \sum_\gamma \psi_{i\gamma}^\dagger \psi_{i\gamma}$, and the bond operator along the nearest-neighbor bond in $\hat{x}$ direction is defined as $B_i = \sum_\alpha (\psi_{i,\alpha}^\dagger e^{i\phi_{i,i+\hat{x}}} \psi_{i+\hat{x},\alpha} + \text{h.c.})$. $\alpha, \beta \in \{\uparrow, \downarrow\}$ are spin flavor indices. We have written the spin operators as antisymmetric self-conjugate representation [1]. They are related to the normal spin operators definition as $S^{\uparrow\downarrow} = S^+, S^{\downarrow\uparrow} = S^-, S^{\uparrow\uparrow} = S^z, S^{\downarrow\downarrow} = -S^z$.

Other than the gauge invariant fermionic observables, we also measure the correlation of the magnetic flux operator $\theta_\square := \text{curl}\phi = \sum_{ij \in \text{edge}(\square)} \phi_{ij}$, where $\phi_{ij} = -\phi ji$ and the summation is along the positive directions shown in Fig. 1. We focus on the imaginary time dynamics. Since, in the static limit ($J = 0$) of the QED$_3$, the ground energy magnetic configuration is known to be $\pi$-flux on a square lattice [61]. When $J$ is turned on, the fluctuation of magnetic flux is centered also around $\pi$-flux. So, to study the correlation of the magnetic fluxes, we measure the correlation of its sine value instead, in order to have a clean background, i.e. $\langle \sin \theta_\square \rangle = 0$. We define

$$C_{\text{flux}}(\tau) = \frac{1}{V_s} \sum_\square \langle \sin \theta_\square(\tau) \sin \theta_\square(0) \rangle. \qquad (6)$$

Next, the CFT operator scaling dimension are readily extracted from the correlation functions. As introduced before, the DSL with non-compact QED$_3$ has been analyzed extensively and is believed to describe a scale-invariant CFT. Consequently, power law behavior is expected for all gauge-invariant

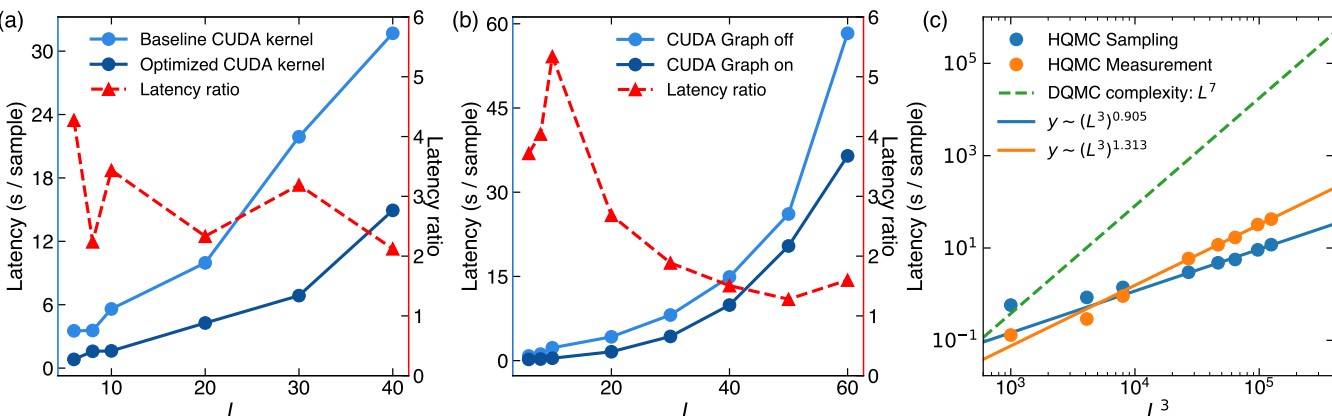

FIG. 2. **The speedup achieved by customized CUDA kernel and CUDA Graph, and the computational complexity of HQMC.** **(a)** shows the speedup achieved by customized CUDA kernel for different system sizes $L$, with $V_s = L^2$ and $N_\tau = 10L$. The blue axis on the left represents the latency of the Monte Carlo sweep which is average time cost per generated sample. The light blue line represents the baseline performance, where the PyTorch sparse-matrix-vector multiplication is applied to compute $M^\dagger M v$ and $\tilde{O}^{-1} v$. The dark blue line is the latency of the customized CUDA kernel, where the computations $M^\dagger M v$ and $\tilde{O}^{-1} v$ are replaced with optimized CUDA kernel. The red dashed line is the latency ratio, which shows the speedup between the two implementations for each system size. **(b)** shows the speedup achieved by CUDA Graph, for different system sizes $L$. The light blue line represents the baseline performance with CUDA Graph turned off, while the dark blue line is the latency when CUDA Graph is turned on. The red dashed line is the latency ratio, which shows the speedup between the two implementations for each system size. **(c)** shows the scaling behavior of the HQMC sample generation latency and HQMC measurement latency using stochastic estimation (SE), as functions of $N_\tau V_s \propto L^3$, where $N_\tau = \beta/\Delta_\tau = 10L$. The latency (second per sample) is defined as the time spent to generate a gauge sample, or the time to compute a fermionic observable from a gauge sample for SE (the SE latency here is tested on spin-spin correlation using $N_{\text{rv}} = 40$). The blue and orange line are the linear fitting of five large-size points. They show that the HQMC algorithm has a time complexity of $O((L^3)^{0.9})$, while the stochastic estimation has a time complexity of $O((L^3)^{1.3})$. As a contrast, the DQMC has a time complexity of $O(N_\tau V_s^3) = O(L^7)$, which is plotted as the green dashed line. The starting point of the green dashed line is 0.38 second per sweep at $L^3 = 10^3$, obtained from DQMC benchmark.

observables. In our setting of $N_f = 2$, the CFT is expected to have emergent $SU(4)$ symmetry (with two cones in the BZ), one expects the scaling dimension of the fermion bilinears or mass terms to be the following: for a singlet $\Delta_s \approx 2.4$ and for a set of adjoints $\Delta_{\text{adj}} \in (1.75, 1.95)$, where there are 15 of them ranging over the $SU(4)$ generators. These scaling dimensions are analytically obtained by the $\epsilon$-expansion from $d = 4 - 2\epsilon$ to $d = 3$, in one of the most up-to-date computations[17]. We note that, the spin operator and the bond operator mentioned above in Eqs. (4) and (5) are among the 15 adjoint fermion bilinear terms and consequently they are expected to share the same power-law ($\sim 1/r^{2\Delta_{\text{adj}}}$) when we compute their correlation functions [3, 6] and it is the indeed the case as we show in the next section.

Moreover, in the $U(1)$ DSL phase, one expects that the flux-flux correlations is dictated by the scaling dimension of the conserved current $\Delta_J = 2$, which leads to the long-time correlations of $C_{\text{flux}}(\tau) \sim \tau^{-4}$. In contrast, for a non-compact $U(1)$ pure free-photon theory without fermion coupling, which is non-conformal, the flux correlation decays with a different power, $C_{\text{flux}}(\tau) \sim \tau^{-3}$ [62]. From our HQMC simulation, the $\tau^{-4}$ behavior of flux correlation is clearly seen at large space-time volume.

**Numerical results**

Fig. 3(a) shows the log plot of the spin-spin correlation

$C_S^{\uparrow\downarrow}(r, 0)$ as a function of real space distance $r$ at $J = 1.25, K = 1$, for various lattice sizes. As the system size increases from $L = 12$ all the way to $L = 60$, the spin correlation curves monotonically unroll, and approach a straight envelope line at large distances. This indicates a power law decay behavior. By manually fitting this envelop curve shown as the dark solid line, we obtain the exponent of the power law decay to be around 3.8. This agrees with the scaling dimension of the fermion bilinear operators $2\Delta_{\text{adj}} \in (3.5, 3.9)$ predicted by the CFT, where the fermion bilinear operators serve as the adjoint representation of the emergent $SU(4)$ symmetry. We note that at around $1 \times 10^{-5}$ in Fig. 3(a), the data fall below the noise level. So the manual fitting is performed mainly with the few large-distance data $\gtrsim 10^{-5}$ at large system sizes. In Fig. 3(a), we also plot the DQMC results for several small lattice sizes to benchmark the HQMC results, and they well agree with each other. These results are complemented by the benchmark data in Sec.II of the Supplemental Materials (SM) [63].

Fig. 3(b) shows log plot of the bond-bond correlation $C_B(r, 0)$ as a function of real space distance $r$ at $J = 1.25, K = 1$, for various lattice sizes. The data of the bond correlation generally have a larger variance than the spin correlation function shown in Fig. 3(a), which is due to the explicit gauge field dependence in the bond operator $B_i$. But as the system sizes increases to beyond $L \gtrsim 30$, the decaying behavior of the bond-bond correlation tends to be stable around a straight

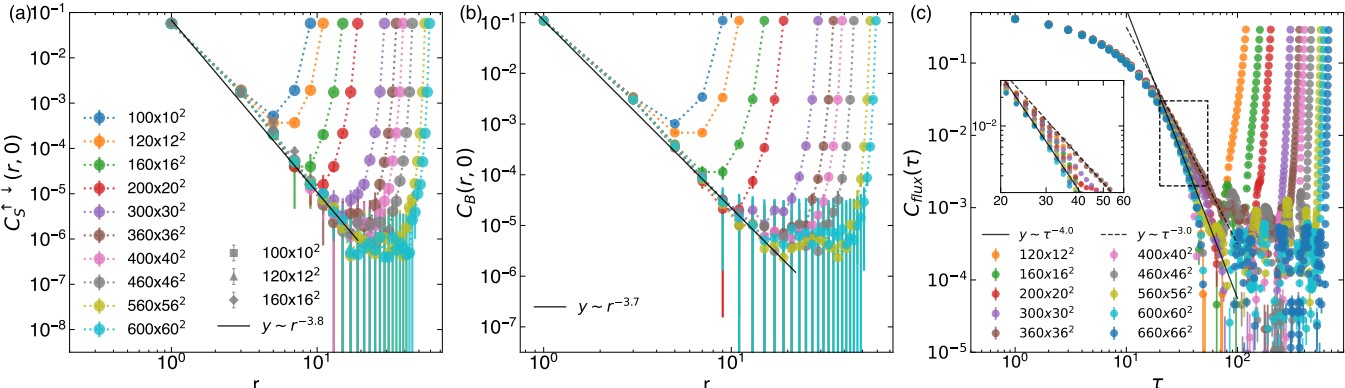

FIG. 3. **The correlation functions for non-compact QED$_3$ with $K = 1$ and $J = 1.25$.** **(a)** shows the log plot of the spin-spin correlation $C_S^{\uparrow\downarrow}(r,0)$ as a function of distance $r$, for various cubic lattice sizes shown as $N_\tau \times L^2$ in the legend. Notice that $C_S^{\uparrow\downarrow} = C_S^{\downarrow\uparrow}$, so we only need to measure $C_S^{\uparrow\downarrow}$ here. To avoid the even-odd oscillation in the finite-size data, we only plot the data of the distance $r$ = odd points. The gray square, triangular and diamond dots show the results computed from the DQMC to benchmark the HQMC result of the corresponding lattice sizes. At the large distance and the large system sizes, the data are manually fitted by the black solid line, which displays a power law decay as $1/r^{3.8}$. **(b)** shows the log plot of the bond-bond correlation (along $\hat{x}$ direction) as a function of distance $r$. The legend is the same as that in (a). At the large distance and the large system sizes, the manually fit line of the bond-bond correlation also displays a power law decay as $1/r^{3.7}$. **(c)** shows the log plot of the flux-flux correlation as a function of imaginary time $\tau$. At the large distance and the large system sizes, the black solid line shows the manual fitting which displays a power law decay as $\tau^{-4.0}$. The black dashed line shows the $\tau^{-3.0}$ power law behavior for comparison.

line, which again indicates a power law decay. The manual linear fitting shown as the dark solid line yields a power of around 3.7, which again agrees with the scaling dimension of the fermion bilinear operators $2\Delta_{\text{adj}} \in (3.5, 3.9)$.

As analyzed in the above section and shown in Figs. 3(a) and 3(b), the fact that both spin and bond correlation functions display power law decay with powers consistent with the scaling dimension of the CFT fermion bilinear operators supports the existence of the $U(1)$ DSL phase in non-compact QED$_3$ at $J = 1.25$. This also agrees with a recent DQMC study on the same model [62].

Fig. 3(c) shows the log plot of the magnetic flux correlation function $C_{\text{flux}}(\tau)$ as a function of imaginary time $\tau$ for various lattice sizes, at $J = 1.25, K = 1$. It can be seen that, as system size increases, the curves become increasingly linear before entering the noise level at around $2 \times 10^{-3}$. Such linear tendency is manually fitted with the dark solid line, which yields a power of around 4.0. In contrast, the black dashed line shows the $\tau^{-3.0}$ behavior, which originates from the non-conformal flux-flux correlation in a $U(1)$ gauge field theory [62]. This is clearly distinct from the $\tau^{-4.0}$ behavior originating from the conserved current's scaling dimension $\Delta_J = 2$, as mentioned above. Thus, the power law decay in the flux correlation function with power 4.0 also indicates the existence of $U(1)$ DSL phase in $J = 1.25$ in the non-compact QED$_3$.

## DISCUSSION

In this work, we developed a scalable GPU-accelerated hybrid quantum Monte Carlo (HQMC) algorithm for simulating $U(1)$ gauge field coupled to fermions in (2+1)d. We make the following three major advances in the computational technique. First, we constructed a problem-specific preconditioner that enables rapid convergence of the conjugate gradient solver. Second, we designed customized CUDA kernels that exploit the sparse structure of the matrix-vector operations, and utilizes the GPU local caching accordingly. Third, we integrated CUDA graph acceleration to minimize kernel launch overhead and maximize GPU utilization. These optimizations collectively allow us to reach a linear scaling in computational complexity with respect to space-time volume, i.e. $O(N_\tau V_s)$, and to simulate unprecedentedly large system sizes with ease.

Beyond the computational advances, our large-scale simulations have enabled us to investigate the non-compact QED$_3$ with better scrutiny. By systematically analyzing the spin-spin and bond-bond correlation functions, we extract the scaling dimensions of the corresponding fermion bilinear operators. Our results demonstrate that both observables exhibit power-law decay with exponents consistent with scaling dimension of the adjoint representation bilinear terms of the emergent SU(4) symmetry computed from field-theoretical approach. Furthermore, the flux-flux correlation function reveals a $\tau^{-4}$ decay at long times, reflecting the scaling dimension $\Delta_J = 2$ of the conserved current. These results provide direct evidence for the conformal nature of the $U(1)$ Dirac spin liquid phase in the non-compact QED$_3$ model.

Since the HQMC gives the full space-time correlation of

the fermion bilinear operators, one immediate future direction is to investigate the spectral information inside the $U(1)$ DSL phase from the stochastic analytic continuation of the HQMC data [62]. As we have both large $N_\tau$ and $V_s$, the momentum and energy resolution of the obtained spectra (for example the magnetic spectra from spin-spin correlation function) will be much higher than those from the DQMC [62]. In this way, the weight distribution and the momentum dependence of the spinon continuum spectra of $U(1)$ DSL can be revealed, and hopefully connection with experiment results on triangular lattice antiferromagnets YbZn$_2$GaO$_5$ [36, 37] and the $A$-YbSe$_2$ delafossites [38, 39] and kagomé antiferromagnet YCu$_3$(OD)$_6$Br$_2$[Br$_{1-x}$(OD)$_x$] [40–46] can be made.

Moreover, by adding magnetic field to the QED$_3$ model, one can investigate how the flavor chemical potential will change the $U(1)$ DSL phase and understand the emergent gauge flux in magnetized Dirac spin liquids. For example, recent DQMC work [62] and analytic works [64, 65] have explored the various symmetry-breaking phases out of $U(1)$ DSL under magnetic field. In particular, a novel chiral flux phase, where the system spontaneously generates a net emergent orbital magnetic flux (deviated from the $\pi$-flux at zero field case) has been seen from smaller size DQMC simulation. This spontaneous orbital magnetic flux changes the behavior of the spinons, and make them form the relativistic Landau levels — energy levels normally seen in electrons moving in a magnetic field. Remarkably, the system also exhibits a gapless photon-like mode, a signature of the coherent emergent gauge field, which can be potentially detected in inelastic neutron scattering experiments [62]. Our GPU-accelerated HQMC algorithm can provide systematic results of such interesting phase and rich diagram, as a function of magnetic field, gauge field fluctuations, as well as the temperature axes. One possible direction is to investigate the effect of different filling of the spinon Landau levels in the magnetized DSL phase [66], and establish possible connections to experimentally observed unusual magnetic oscillations in the YCu$_3$(OD)$_6$Br$_2$[Br$_{1-x}$(OD)$_x$] kagomé antiferromagnet [46].

Taken together, our work establishes a robust and efficient computational framework for exploring the physics of strongly coupled quantum field theories at large scales, and provides new quantitative insights into scaling dimensions of the operators in the $U(1)$ Dirac spin liquid. This technique opens an avenue to future investigations of phase transitions and symmetry-breaking phenomena in related models, and hopefully, making closer connection with the active on-going experimental efforts in finding the $U(1)$ DSL state in quantum magnets and the theoretical efforst in understanding the strongly coupled conformal field theory in 2+1 dimensions in high-energy physics.

## METHODS

### Model and partition function

As introduced above in the Results section, the model we study consists of a fermionic Lagrangian $L_F$, and a gauge field Lagrangian $L_B$. And we consider two $U(1)$ gauge field Lagrangians, namely compact $U(1)$ gauge field $L_B^c$ and non-compact $U(1)$ gauge field $L_B^{nc}$ as introduced in the Results section. The discretized version of curl $\phi$ for a spatial plaquette $\square$[12, 55] , which is also seen as magnetic flux through the plaquette, is defined as $\theta_\square^{xy} = \text{curl } \phi = \sum_{ij \in \text{edge}(\square)} \phi_{ij}$, where $ij$ is along the positive direction, and $\phi_{ij} = -\phi_{ji}$, as illustrated in the bottom right cube in Fig. 1. Similarly, in the cubic space-time lattice, the curl of a temporal plaquette, i.e. $\theta^{\tau x}$ or $\theta^{\tau y}$, can also be defined. If we fix the gauge field of temporal bonds to be zero, then the curl of a temporal plaquette is $\pm(\phi_{ij,\tau+1} - \phi_{ij,\tau})$, which consists of the $\tau$-derivative term in the gauge field Lagrangians. It is then clear that $L_B^c$ has $2\pi$-periodicity in the magnetic flux: $\theta \to \theta + 2n\pi, n \in \mathbb{Z}$. Thus, $L_B^c$ describes a compact $U(1)$ gauge field theory in the sense that the magnetic flux is physical only in the range of $(-\pi, \pi]$, while the other values of $\theta$ are unphysical and need to be wound into the $(-\pi, \pi]$ interval. Due to $2\pi$-periodicity (the fermionic Lagrangian $L_F$ also has $2\pi$-periodicity by definition [61]), the compact QED$_3$ is known to host magnetic monopoles, which contains distinct physics from a non-compact theory e.g. $L_B^{nc}$ in Eq. (2), where the first term breaks the $2\pi$-periodicity. For example, the proliferation of magnetic monopoles in compact QED$_3$ will lead to the emergence of a mass gap in the gauge field, through the dual superconductor mechanism [18, 67, 68], while the non-compact QED$_3$ is generally believed to remain deconfined. We therefore focus on the results on non-compact QED$_3$ in the main text and leave the compact QED$_3$ results in the Supplemental Materials (SM) [63]. Although from the algorithmic point of view, there is no difference in their implementation and performance of the corresponding simulations.

In our numerical simulation, the quadratic fermion $\psi$ can be traced out, yielding

$$\text{Tr}_\psi \left[ e^{-S_f} \right] = \prod_\alpha \left[ \det \left( \mathbf{1} + \prod_{\tau=1}^{N_\tau} B_{\tau,\alpha} \right) \right] = \prod_\alpha \det M_\alpha, \quad (7)$$

where $B_{\tau,\alpha} = e^{V_{\tau,\alpha}}$, and $V_{\tau,\alpha}$ is the exponential of fermion-gauge coupling matrix, whose elements are determined by gauge field $\Delta_\tau e^{i\phi_{ij,\tau}}$. The matrix

$$M_\alpha = \begin{pmatrix} \mathbf{1} & 0 & 0 & \cdots & 0 & B_{N_\tau,\alpha} \\ -B_{1,\alpha} & \mathbf{1} & 0 & \cdots & 0 & 0 \\ 0 & -B_{2,\alpha} & \mathbf{1} & \cdots & 0 & 0 \\ \vdots & \vdots & \vdots & \ddots & \vdots & \vdots \\ 0 & 0 & 0 & \cdots & \mathbf{1} & 0 \\ 0 & 0 & 0 & \cdots & -B_{N_\tau-1,\alpha} & \mathbf{1} \end{pmatrix}. \quad (8)$$

is of size $N_\tau V_s \times N_\tau V_s$. Furthermore, it can be shown [6] that $\det M_\alpha \in \mathbb{R}$ and $\det M_\uparrow = \det M_\downarrow$, thus the simulation is free from the sign problem. Based on the formula above, the partition function can then be expressed as

$$Z = \int [\delta\phi] \, e^{-S_B(\phi)} \prod_\alpha \det M_\alpha(\phi), \quad (9)$$

where $S_B(\phi)$ can be either compact or non-compact bosonic gauge action. The Monte Carlo simulation will then be applied to sample the gauge field $\phi$ and explore the Hilbert space defined by the partition function in Eq. (9), which includes the fermion determinant computation for each $\phi$ configuration.

**HQMC algorithm**

In this subsection, we will briefly introduce the HQMC algorithm applied to simulate the partition function Eq. (9). This algorithm originates from lattice quantum chromodynamics simulation [49], and later finds its application in a broader class of statistical problems [69]. An application in the Hubbard model setting of condensed matter physics can be found in Ref. [51]. In the most recent progress, the method is significantly accelerated with the help of GPU in the study of spin-fermion models and the associated non-Fermi-liquid and quantum critical metallic states [53, 54].

To sample the configurational space of Eq . (9), but avoiding the cubic complexity in computing $\det M$, a commonly used technique in HQMC is to introduce pseudofermion fields $\eta$ and convert the determinant into the exponential,

$$Z = \int [\delta\phi\delta\eta] \, e^{-S_B(\phi)-\eta^\dagger [M^\dagger(\phi)M(\phi)]^{-1}\eta}. \quad (10)$$

Here, the Gaussian integral of the complex fields $\eta \in \mathbb{C}^{V_s N_\tau}$ has been conversely used, and the identity $\det M_\uparrow = \det M_\downarrow \in \mathbb{R}$ is assumed, and proved in Ref. [6]. Thus, the fermionic flavor $\alpha$ is omitted.

Next, we introduce the Hamiltonian dynamics, which provides a mechanism to evolve the gauge field continuously along an equal-energy trajectory. Thus the phase space is explored effectively with high acceptance rate. We further add a momentum field $p_{ij,\tau}$ canonical to the gauge field $\phi_{ij,\tau}$:

$$Z = \int [\delta\phi\delta p\delta\eta] \, e^{-H(\phi,p,\eta)}, \quad (11)$$

where

$$H(\phi,p,\eta) := S_B(\phi) + \sum_{ij,\tau} p_{ij,\tau}^2 + \eta^\dagger \left[(M^\dagger(\phi)M(\phi))\right]^{-1}\eta. \quad (12)$$

In the partition function Eq. (11), variables $p$ and $R := [M^\dagger(\phi)]^{-1}\eta$ obey the Gaussian distribution. Then, the HQMC algorithm that samples the partition function Eq. (11) can be readily introduced as the following steps. For a given initial bosonic field configuration $\phi_0 \in \mathbb{R}^{2V_s N_\tau}$,

(i) sample $p_0 \in \mathbb{R}^{2V_s N_\tau}$ and $R \in \mathbb{C}^{V_s N_\tau}$ from the Gaussian distribution; solve for $\eta = M^\dagger(\phi_0)R$;

(ii) evolve $(\phi_0, p_0)$ by the Hamiltonian dynamics described by the differential equations:

$$\dot{p}_{ij,\tau} = -\frac{\partial H}{\partial \phi_{ij,\tau}} \quad (13)$$

$$\dot{\phi}_{ij,\tau} = \frac{\partial H}{\partial p_{ij,\tau}} = 2p_{ij,\tau}, \quad (14)$$

which will yield the proposal variables $(\phi_t, p_t)$; note that the $t$ here is the time parameter for the differential equation;

(iii) check $(\phi_t, p_t)$ with the Metropolis-Hastings rule to get the acceptance ratio:

$$r_{\text{MH}} = \min\left(1, e^{H(\phi_0,p_0,\eta)-H(\phi_t,p_t,\eta)}\right); \quad (15)$$

output $(\phi_t, p_t)$ if accepted and $(\phi_0, p_0)$ otherwise;

(iv) set $\phi_0$ as the output $\phi$ and go to the next iteration starting from (i).

Some further clarifications of this algorithm are in order.

First, the Hamiltonian dynamics has the property that the phase space volume is conserved, according to the Liouville theorem [51], which ensures that $H(\phi_0, p_0, \eta) = H(\phi_t, p_t, \eta)$ and that the proposal variables $(\phi_t, p_t)$ are almost always accepted, if the differential equation is solved exactly.

Second, we apply the well-known Leapfrog method [51] to solve the differential equation Eq. (14). It has the property of time reversibility, which, together with the Metropolis-Hasting rule, guarantees that the Monte Carlo updates fulfill the detailed balance condition [51]. The Leapfrog method is described as follows. At the $n$-th step, to evolve a given $(\phi_{n-1}, p_{n-1})$ for a time step $\Delta t$,

$$p_{n-\frac{1}{2}} = p_{n-1} + F(\phi_{n-1})\frac{\Delta t}{2}$$
$$\phi_n = \phi_{n-1} + 2p_{n-\frac{1}{2}}\Delta t \quad (16)$$
$$p_n = p_{n-\frac{1}{2}} + F(\phi_n)\frac{\Delta t}{2}.$$

Here, the force $F(\phi) := -\frac{\partial H}{\partial \phi}$ consists of both the bosonic force $-\frac{\partial S_B}{\partial \phi}$ and the fermionic force $-\frac{\partial}{\partial \phi}\left\{\eta^\dagger \left[(M^\dagger(\phi)M(\phi))^{-1}\eta\right]\right\}$, the derivation of which is shown in SM Sec. . In this calculation, generically a matrix vector of the form $[M^\dagger M]^{-1}v$ is involved, which is equivalent to solving a linear system $[M^\dagger M]X = v$, and will be calculated using a preconditioned conjugate gradient (pCG) algorithm, which will be explained in section CG Method and Preconditioner below. In this Leapfrog method, the time step size $\Delta t$ and the number of leapfrog steps $N_{\text{leapfrog}}$ in a Monte Carlo sweep are two tunable parameters. Larger $N_{\text{leapfrog}}$ benefits from larger change in the proposed configuration, but suffers from the larger accumulated error in the iterative process. We choose $N_{\text{leapfrog}} = 3$ in our implementation. The time step size $\Delta t$, on the other hand, is adaptively tuned to maintain a moderate acceptance rate.

Third, since the update scheme in HQMC is based on solving the Hamiltonian differential equation, it is naturally continuous. As a result, the random walk in HQMC is limited to a continuous region the configuration space. But in the QED$_3$ model studied in this paper, there are two non-continuous regions, namely those with $(\det M_\uparrow, \det M_\downarrow) = (+, +)$ and

$(\det M_\uparrow, \det M_\downarrow) = (-, -)$. They are separated by the gauge configurations where $\det M \approx 0$, which themselves have diminishing contribution in the partition function Eq. (9), but prevents the HQMC update from going from one region to the other. This leads to the ergodicity issue, shared by the application of HQMC on other models [51] as well. One common solution is to introduce another legitimate update that is ergodic. Then the combination of the multiple updates will cure the ergodicity issue in HQMC [69]. In Ref. [51], the solution of this issue is to decouple the Hamiltonian into two channels and then to complexify the configuration space, which is more problem-specific. In the QED$_3$ studied here, we find that in the small $J$ region, which is the $U(1)$ DSL phase region and the focus of this paper, the ergodicity issue is numerically negligible. So we directly apply the HQMC algorithm in this region. The validity is benchmarked by the DQMC algorithm and is shown in Sec. II of SM [63].

**Stochastic estimation of fermionic observables**

For a given bosonic configuration $\phi$, the fermionic observables are easily computed by using the standard stochastic estimation procedure [49, 70]. More specifically, the single-particle Green's function, $G_{ij} = \langle c_i(\tau) c_j^\dagger(0)\rangle$, which is obtained from the inversion of the $M$ matrix, $M^{-1}$. The direct inversion of $M$ is numerically expensive and scales as $O((V_s N_\tau)^3)$. Alternatively, the standard stochastic procedure to compute $M^{-1}$, as an iterative method, is usually more efficient. In this procedure, first sample a random vector $\xi \in \mathbb{C}^{V_s N_\tau}$, whose entries are i.i.d. and have the orthogonality property $\mathbb{E}_\xi \xi_i^* \xi_j = \delta_{ij}$. Then, $\sum_l \left[M^{-1}\right]_{il} \xi_l \xi_j^*$ will give an unbiased estimation of $\left[M^{-1}\right]_{ij}$, since

$$\mathbb{E}_\xi \sum_l \left[M^{-1}\right]_{il} \xi_l \xi_j^* = \left[M^{-1}\right]_{ij}.$$

Here, the random vector can be drawn from any distribution that satisfies $\mathbb{E}_\xi \xi_i^* \xi_j = \delta_{ij}$, including Gaussian distribution, Rademacher distribution, etc [70]. In practice, we compute $\sum_l \left[(M^\dagger M)^{-1}\right]_{il} \left[M^\dagger \xi\right]_l \xi_j^*$, instead, where $R := M^\dagger \xi$ is numerically cheap to compute and $\left[M^\dagger M\right]^{-1} R$ can be computed generically using a pCG algorithm. In this way, the Green's function can be obtained, and other more complicated correlation functions can be further obtained from Green's function via Wick decomposition.

Note that in practice, when numerically computing the Green's functions, we utilize the translational invariance and take the average over space and imaginary time, i.e. $C(i, j) = C(0, \Delta)$. Here, $i, j$ are site indices of the cubic spatial-temporal lattice, $\Delta := j - i$, and $C(i, j)$ is either two-point, four-point or eight-point Green's functions. Apparently, naively taking the average

$$C(\Delta) = \frac{1}{N} \sum_i C(i, i + \Delta) \tag{17}$$

has a time complexity of $O(N^2)$, where $N = V_s N_\tau$. Here, we follow the Ref. [70], and apply the fast Fourier transformation (FFT) instead of directly eevaluating this average.

As introduced above, for a given bosonic field configuration, and a random vector $\xi$, an unbiased estimation of the two-point Green's function is $\hat{G}_{ij} = [G\xi]_i \xi_j^*$. Thus, an unbiased estimation of any correlation function $C(i, j)$, either the four-point Green's function or the eight-point Green's function, can be uniformly written in the form of

$$\hat{C}(i, j) = a_i b_j, \tag{18}$$

where $a$ and $b$ are two vectors. For example, if $C(i, j) = G_{i,j} G_{i,j}$, then for an unbiased estimation of $\hat{C}(i, j)$, $a_i = (G\xi)_i (G\xi')_i$ and $b_j = \xi_j^* \xi_j^{*\prime}$. Then, the average of $\hat{C}(i, i + \Delta)$ takes the convolution form, and FFT is readily applied,

$$\hat{C}(\Delta) = \frac{1}{N} \sum_i a_i b_{i+\Delta} = \mathcal{F}^{-1}[\mathcal{F}[a]_{-k} \cdot \mathcal{F}[b]_k](\Delta), \tag{19}$$

where $\mathcal{F}$ is FFT operation. Then the time complexity is reduced to $O(N \log N)$, with $N = V_s N_\tau$. Moreover, as introduced in Ref. [70], each random vector among $N_{\text{rv}}$ random vectors can serve as either $\xi$ or $\xi'$ in the example above. Thus, totally $f(N_{\text{rv}}) = \binom{N_{\text{rv}}}{2}$ random vectors can be effectively used in the stochastic estimation. In our benchmark, $N_{\text{rv}} = 40$ is sufficient to reduce the error of fermionic bilinear's correlation function to a moderate level.

**CG method and preconditioner**

As introduced above, the key step in reducing the cubic complexity of computing the determinant $\det M$ or the inversion $M^{-1}$ is to convert it to the evaluation of $\left[M^\dagger M\right]^{-1} v$, which is equivalent to solving a linear system $OX = v$. Here, $O = M^\dagger M$ is a Hermitian positive semi-definite matrix; thus, the equation can be efficiently solved by the iterative preconditioned conjugate gradient (pCG) method [51].

The pCG iteration can be summarized as follows. Given the linear equation $OX = v$ and the preconditioner $\tilde{O}^{-1}$, initialize the vectors $X_0 = \mathbf{0}$, $r_0 = v - OX_0$, $d_0 = r_0$ and the scalar $\zeta_0 = \langle r_0, \tilde{O}^{-1} r_0\rangle / \langle d_0, O d_0\rangle$. Then, for $n = 0, 1, 2, \ldots$ until convergence,

$$
\begin{aligned}
X_{n+1} &= X_n + \zeta_n d_n \\
r_{n+1} &= r_n - \zeta_n O d_n \\
\rho_n &= \langle r_{n+1}, \tilde{O}^{-1} r_{n+1}\rangle / \langle r_n, \tilde{O}^{-1} r_n\rangle \\
d_{n+1} &= \tilde{O}^{-1} r_{n+1} + \rho_n d_n \\
\zeta_{n+1} &= \langle r_{n+1}, \tilde{O}^{-1} r_{n+1}\rangle / \langle d_{n+1}, O d_{n+1}\rangle,
\end{aligned} \tag{20}
$$

where $\langle \cdot, \cdot \rangle$ is the inner product. The time complexity of this algorithm is decided by the matrix-vector multiplication time complexity in $\tilde{O}^{-1} r$ and $Od$, and the number of pCG iterations.

In large-scale computations of pCG algorithm, the application of preconditioner is crucial in reducing the number of iterations required to reach convergence. However, the choice

of preconditioner is usually problem-specific; there is not a universal way of obtaining a proper preconditioner for any linear system [51]. In principle, a good choice of preconditioner needs to balance the accuracy and efficiency. The accuracy is to measure how well the preconditioner approximates the inverse of the $O$ matrix in $OX = v$, i.e., the preconditioner $\tilde{O}^{-1} \approx O^{-1}$. In the most ideal case, if $\tilde{O}^{-1} = O^{-1}$, then it would require only one iteration to reach convergence and obtain $X = O^{-1}v$. But if the preconditioner were too complex to obtain or to utilize in pCG iterations, then the pCG process would not be accelerated.

In our case, the preconditioner $\tilde{O}^{-1}$ is obtained in the following procedure. First, for a given matrix $O = M^\dagger(\phi)M(\phi)$ for some generic fixed bosonic field $\phi$, we apply the incomplete Cholesky decomposition to obtain a lower triangular matrix $L(\phi)$, such that $O \approx L^\dagger(\phi)L(\phi)$. Then, we apply the Neumann series expansion to approximate the inverse of $L$ as

$$L^{-1} \approx 1 + (1 - L) + (1 - L)^2 + \ldots, \tag{21}$$

for 20 iterations. Finally, we truncate the entries with values smaller than $1 \times 10^{-3}$, and obtain the preconditioner $\tilde{O}^{-1} = L^{-1}(\phi)L^{-\dagger}(\phi)$. Here, all computations, including incomplete Cholesky decomposition and triangular matrix multiplications, have time complexity $O((V_s N_\tau)^2)$, which is numerically affordable. Most importantly, we find that the preconditioner in our problem is endowed with good generalizibility, in the sense that the preconditioner obtained at a generic bosonic field $\phi$ is applicable to accelerate the pCG iteration at any other bosonic fields throughout the Monte Carlo simulation. So in practice, we just use $\pi$-flux bosonic configuration to compute the preconditioner once and apply it globally throughout the MC simulation.

In essence, such generalizability in the preconditioner is because in our problem, as shown in Eq. (8), the $M$ matrix is very weakly modulated by the bosonic field, as long as $\Delta_\tau$ is small enough. Our computation shows that as long as $\Delta_\tau \leq 0.1$, the pre-computed preconditioner has good generalizability. But if $\Delta_\tau \gtrsim 0.12$, then the pre-computed global preconditioner will fail and have a negative impact on the convergence of pCG iterations.

Due to the same reason, we find another fold of the generalizability in our preconditioner, which addresses the exploding memory consumption issue when the size goes to up $N_\tau \times L^2 = 660 \times 66^2$. We find that as system size increases, not only does the sparsity pattern of the preconditioner remain the same, but also the entry values at the bulk of the preconditioner simply repeat. So we can directly obtain the preconditioner without any computation.

Fig. 4 shows pCG convergence error $\varepsilon := \frac{||v - OX_n||}{||v||}$ for different fixed numbers of iterations for various sizes. This is dual to a more natural description where people measure the number of iterations required to converge to fixed error instead. Here, we take the former description because of the static nature of pCG iteration implemented with CUDA Graph, as introduced in section CUDA Graph Acceleration

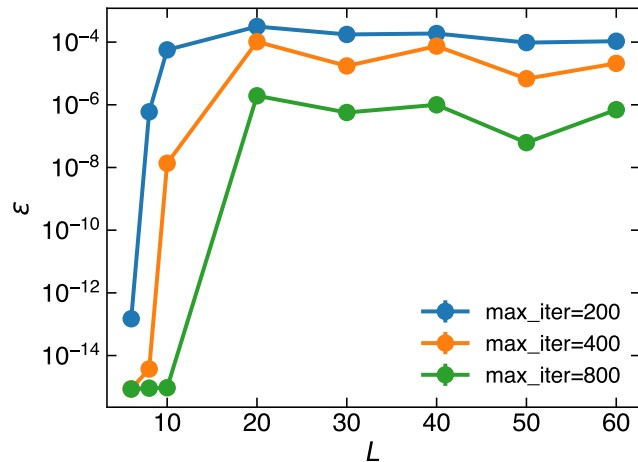

FIG. 4. **The convergence of the preconditioned-CG versus size.** The semi-log plot of the pCG convergence error $\varepsilon := \frac{||v - OX_n||}{||v||}$ as a function of lattice linear size $L$ for various fixed max number of pCG iterations. In the benchmark, the pCG convergence error are measured and averaged in a MC process, where we set $J = 1.25$, $\Delta_\tau = 0.1$. The total cubic lattice sizes for each $L$ is $N_\tau V_s = 10L^3$. At large lattice sizes, the convergence error levels off, which indicates that the number of pCG iterations required to converge has a constant complexity.

below. This plot shows that for each fixed number of iteration, the convergence error does not increase with system size. This indicates that the number of pCG iterations required to converge to a given precision, i.e. the iteration number complexity, is constant.

### Customized CUDA kernel for matrix-vector multiplication

The mathematical operations with the highest cost in the pCG iterations are the two matrix-vector multiplications, namely $M^\dagger Mv$ and $\tilde{O}^{-1}v$ multiplication. It turns out that these two operations can benefit significantly from writing customized CUDA kernels, which effectively exploit the specific sparsity pattern of the matrices.

*Matrix-free representation of $M^\dagger Mv$ multiplication*

A naive implementation of this $M^\dagger Mv$ operation would first compute $M^\dagger M$, and store it as a sparse matrix, and then time it with the vector $v$. Generally speaking, the multiplication of a sparse matrix $M^\dagger M$ with a vector $v$ has the time complexity of $O(nnz)$ where $nnz$ is the number of non-zero elements in the matrix $M^\dagger M$, which can reach the scale of $O(N^2)$ when the matrix is not sparse enough. However, instead of the sparse-matrix representation, the definition of the $M$ matrix in Eq. (8) admits a matrix-free representation in the matrix-vector multiplication, which is more efficient [71]. This technique helps achieve a $2.9 - 3.8$ times speedup on top of the baseline implementation, as will be shown next.

In the definition of the $M$ matrix in Eq. (8), we have $B_\tau = e^{V_\tau}$, with $[V_\tau]_{ij} = \Delta_\tau e^{i\phi_{ij,\tau}}$. Then, due to the geometry of the square lattice, one can symmetrically decompose the $V_\tau$ matrix into four families, as denoted as the four bond colors in Fig. 1:

$$B \approx e^{V_4/2} e^{V_3/2} e^{V_2/2} e^{V_1/2} e^{V_1/2} e^{V_2/2} e^{V_3/2} e^{V_4/2} \qquad (22)$$

($\tau$ indices are omitted), which is also known as the symmetrized checkerboard decomposition [6]. This approximation amounts to ignoring the non-commutativity between different families of $V$ matrices, and thus has the same error contribution as the Trotter error. But within a bond family, all hopping terms commute with each other. Therefore, the matrix $V_{\mathrm{fam}_n}$ can be written as a direct sum of $2 \times 2$ matrices:

$$e^{V_{\mathrm{fam}_n}} = e^{\Delta\tau} \bigoplus_{ij \in \mathrm{fam}_n} T_{ij}, \qquad (23)$$

where $T_{ij} = \begin{bmatrix} 0 & e^{i\phi_{ij}} \\ e^{-i\phi_{ij}} & 0 \end{bmatrix}$, which has property of $T_{ij}^2 = \mathbf{1}$. Then within a bond family $\mathrm{fam}_n$, the exponentiation can be readily evaluated:

$$e^{V_{\mathrm{fam}_n}} = \bigoplus_{ij \in \mathrm{fam}_n} e^{\Delta_\tau T_{ij}} \qquad (24)$$

$$= \bigoplus_{ij \in \mathrm{fam}_n} \left( \mathbf{1} \cosh \Delta_\tau + T_{ij} \sinh \Delta_\tau \right) \qquad (25)$$

$$= \begin{bmatrix} \ddots & & \\ & \cosh \Delta_\tau & e^{i\phi_{ij}} \sinh \Delta_\tau \\ & e^{-i\phi_{ij}} \sinh \Delta_\tau & \cosh \Delta_\tau \\ & & & \ddots \end{bmatrix}_{\mathrm{fam}_n} . \qquad (26)$$

It is then clear that the $Bv$ multiplication can be written as a composite of serial sparse-matrix-vector multiplication $e^{V_{\mathrm{fam}_n}} v$, which is straightforwardly parallelizable in GPU, and has a time complexity of $O(V_s N_\tau)$. Furthermore, in the series of $u = e^{V_{\mathrm{fam}_n}} v$ matrix-vector multiplication, the intermediate input vector $v$ and output vector $u$ can be put into the shared memory of GPU, which spares the communication cost between the high-bandwidth memory (HBM) and the on-chip static random-access memory (SRAM) [56].

*Preconditioner-vector multiplication $\tilde{O}^{-1}v$*

As shown in the Fig. 5 (a), our preconditioner obtained in the procedure introduced in section CG Method and Preconditioner has a fixed banded sparsity pattern in the $\tau$ dimension, with a bandwidth 6. Here we will show how to exploit such sparsity pattern to accelerate preconditioner-vector multiplication $\tilde{O}^{-1}v$.

A standard way to write a CUDA kernel for such banded sparse matrix-vector multiplication is illustrated in Fig. 5 (b), exemplified here with a bandwidth of 1 [53, 71]. In this illustration, a block of threads is assigned to process the rows of

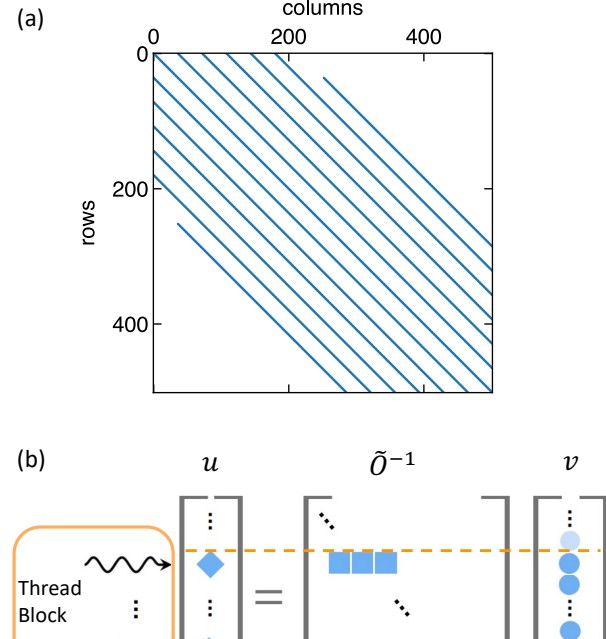

FIG. 5. **Banded preconditioner matrix and its CUDA implementation.** (a) The plot shows the banded sparsity pattern of the preconditioner $\tilde{O}^{-1}$, where the nonzero elements of the first 500 by 500 entries are displayed. The bandwidth is 6. This preconditioner is obtained for a lattice of $N_\tau \times L \times L = 60 \times 6 \times 6$. Two neighboring diagonal lines are separated by a distance of $V_s = L \times L$ which is 36 in this illustration. (b) Schematic of the matrix-free preconditioner-vector multiplication $u = \tilde{O}^{-1}v$, which is illustrated with a banded preconditioner matrix with bandwidth 1. A block of threads are assigned to process the rows of the preconditioner matrix sandwiched by the orange dashed line, with one thread processing one row. The chunk of elements in $v$ involved in this computation are denoted by blue dots, where light blue ones are called the halo elements. In our CUDA kernel implementation, the chunk of $v$ represented by the blue dots are copied into shared memory, which utilizes the local caching of the GPU and reduces the communication overhead.

the matrix, with one thread corresponding to one row of the matrix. In turn, in the matrix-vector multiplication of these rows requires an input of a chunk of the vector $v$ shown as the blue dots in the figure. Apparently, the thread block size is equal to the number of rows being processed and is equal to the vector chunk size minus the number of the so-called halo elements (the light blue dots in Fig. 5 (b)). Thus, the computation is well parallelized. Besides, before the computation, the chunk of the vector is first copied to the shared memory. In this way, the neighboring elements in the vector can be simultaneously read by the neighboring threads. Such local caching has thus been exploited to effectively minimize the communication overhead between HBM and the on-chip SRAM, similar to the $M^\dagger M v$ kernel above. The time complexity of this preconditioner-vector multiplication $\tilde{O}^{-1}v$ is linear

with respect to $N_\tau$ and $V_s$.

*Speedup of customized CUDA kernel*

Fig. 2(a) shows the speedup of the above two customized CUDA kernels over the naive implementations. The baseline for implementing $M^\dagger M v$ and $\tilde{O}^{-1} v$ is to directly call PyTorch sparse-matrix-vector multiplication function, where $M^\dagger M$ and $\tilde{O}^{-1}$ are stored as sparse compressed row storage matrices. The hardware is NVIDIA l40s GPU. Our test shows that the customized CUDA kernel achieves up to 2.9 – 3.8 times speedup across all system sizes.

**CUDA Graph acceleration**

CUDA graph is another important technique that helps achieve significant speedup in the numerical simulation, which is around 2-5 times on moderate lattice sizes (around $L < 30$) and 1-2 times on large system sizes, compared to the baseline implementation. This technique was not introduced until very recently, and has been widely applied in a variety of large-scale GPU-accelerated computations and systems ever since [57, 58].

The principle of the CUDA graph is illustrated in Fig. 6. Without using CUDA graph, if we have a stream of GPU operations to process, generically the operations, implemented as CUDA kernels, will need to be launched one by one in the CPU and then executed in the GPU. As illustrated in Fig. 6, such repetitive launch of CUDA kernels in CPU leads to significant total launching overhead, and also the intermittent execution of kernels causes the GPU to idle between two executions; the total wasted time is comparable to the GPU effective utilization time. But if CUDA graph is applied, what it does differently is to first launch and execute a stream of kernels in a warm-up phase, and record (compile) the computation into a computation graph. Then, in the actual execution phase, the whole computation graph is launched only once and all operations are executed altogether in the GPU. Thus, the speed-up is achieved by reducing the total kernel launching overhead.

However, this technique has the following major limitations that need to be considered in application. The first is that the storage of the entire computation graph increases the GPU memory usage. So, GPU memory capacity can be a bottleneck when scaling up the computation. The second limitation is that the recorded CUDA graph is static, in the sense that once the CUDA graph is recorded, all the subsequent executions of the graph cannot early exit and that the input array must remain the same size and shape at the same memory address, while only its entry values can vary in each CUDA graph launch.

An important consequence of this static graph limitation is that, if the whole pCG iterations are recorded as a CUDA graph, then the number of iterations will be fixed, which cannot be early terminated when the error reaches a fixed precision level. So in our benchmark, instead of fixing the precision level and testing the convergence iteration number, we fix the number of iterations and test the precision.

Despite the limitation, the speedup of this technique is important. Fig. 2(b) shows the latency of a HQMC sweep for various lattice sizes and compares two scenarios between CUDA Graph on and off. In our implementation, we convert the whole fermionic force computation, as well as the stochastic estimation of fermionic Green's function, into CUDA Graph. Fig. 2(b) shows that the execution is around 2-5 times faster for $L < 30$ and 1-2 times faster for larger lattice sizes. The reason that the speedup from CUDA Graph decreases with increased system size, is because as the system size increases, the GPU kernel execution time is increasingly intense. Thus, the kernel launching time becomes relatively small compared to the kernel execution time. The speed up from CUDA Graph acceleration is mainly from the saving of kernel launching latency, so as system size increases, the speedup becomes smaller.

The speedup from CUDA Graph is orthogonal to the speedup obtained from the customized CUDA kernels, thus the speedups are multiplicative.

**Complexity analysis**

Now we give an overview of the complexity analysis of our HQMC. As introduced above, the main iteration of generating a sample in HQMC consists of first $N_{\text{leapfrog}}$ steps of Leapfrog integration as shown in Eq. (16). Within each Leapfrog step, in computing the fermionic force, $N_{\text{pCG}}$ steps of pCG iteration shown in Eq. (20) are required to converge to a given precision. Finally, within each pCG iteration, two matrix-vector multiplications are involved, which have time complexity of $O(N_\tau V_s)$. So the overall time complexity of generating a sample is

$$T_{\text{sample}} = N_{\text{leapfrog}} \cdot N_{\text{pCG}} \cdot O(N_\tau V_s). \qquad (27)$$

As introduced above, $N_{\text{leapfrog}}$ and $N_{\text{pCG}}$ have constant time complexity. So the time complexity of generating a sample is $T_{\text{sample}} = O(N_\tau V_s)$, which is shown as the blue line in Fig. 2(c).

The stochastic estimation for a given sample has similar scaling of time complexity. For a given random vector, a pCG iteration is also called in the stochastic estimation process. So the time complexity is $O(f(N_{\text{rv}}) N_\tau V_s)$, where $N_{\text{rv}}$ is the number of random vectors, and $f(N_{\text{rv}})$ is a polynomial function of $N_{\text{rv}}$, which is fixed to be $\binom{N_{\text{rv}}}{2}$ in this benchmark of fermionic bilinears' correlation functions. Such scaling behavior with respect to $N_\tau V_s$ is shown as the orange line in Fig. 2(c).

As shown in Fig. 2(c), both the sample generation and stochastic estimation display nearly linear scaling behavior with respect to $N_\tau V_s$ as expected, so does the overall time complexity. Moreover, the stochastic estimation has slightly larger scaling power 0.905 than sample generation 1.313, and also larger latency at large system sizes. This is because stochastic estimation needs to simultaneously process $f(N_{\text{rv}})$ input vectors, while sample generation only needs to process one. Thus, the stochastic estimation has much stronger arithmetic intensity, and the latency scales faster with respect to system size. Nevertheless, Fig. 2(c) also shows a green dashed guideline which is the computational complexity of DQMC. At system size of $L^3 \sim 10^5$, the two algorithms already have two orders

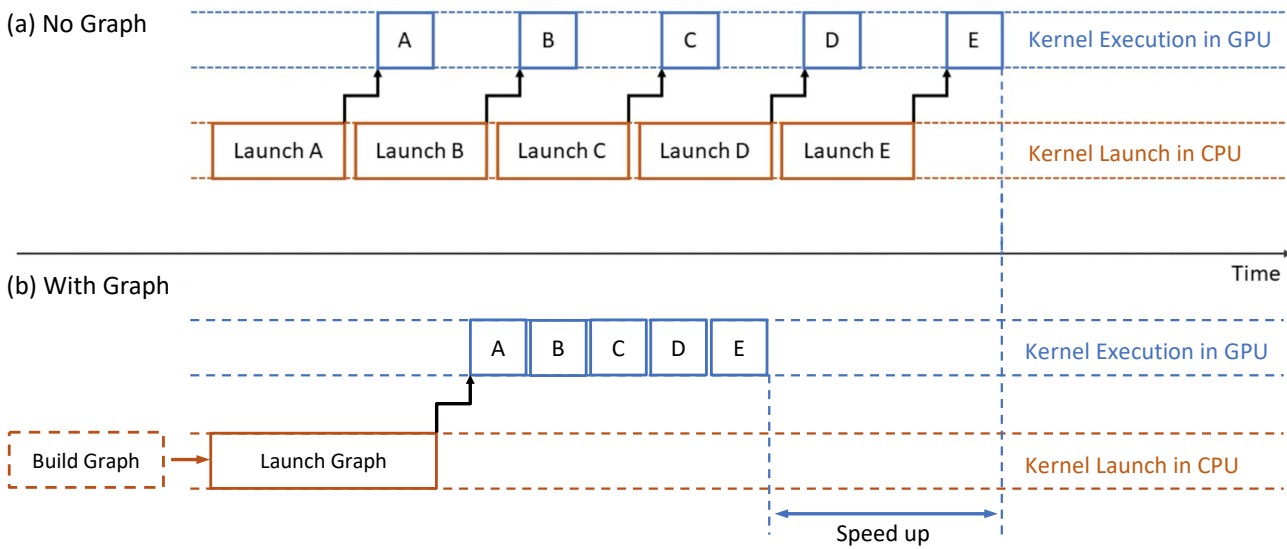

FIG. 6. **Illustration of CUDA Graph acceleration.** (a) The upper half illustrates the naive GPU implementation without using CUDA Graph. The blue boxes are a series of operations (kernels) to perform in GPU, which need to be launched one by one in CPU, represented by the orange rectangles. (b) The lower half shows the mechanism of CUDA Graph acceleration, where the series of operations (kernels) are first compiled together, which happens in the Build Graph phase, and then launched only once in the execution phase. By comparing the wall-clock time between the upper and the lower half, one can see the achieved speedup.

of magnitude latency difference, which shows the advantage of the HQMC.

## Acknowledgments
We thank U.F.P. Seifert, O.A. Starykh and L. Balents for the stimulating discussion on the physics of $U(1)$ DSL and QED$_3$, and the collaboration on a related work [62]. We thank M. Ulybyshev and F. Assaad for inspiring communications on hybrid Monte Carlo. KXF thanks Natalia Perkins for the collaboration in a related project. KXF, CC and ZYM acknowledge the support from the Research Grants Council (RGC) of Hong Kong (Project Nos. AoE/P-701/20, 17309822, HKU C7037-22GF, 17302223, 17301924), the ANR/RGC Joint Research Scheme sponsored by RGC of Hong Kong and French National Research Agency (Project No. A HKU703/22). We thank HPC2021 system under the Information Technology Services and the Blackbody HPC system at the Department of Physics, University of Hong Kong, as well as the Beijing Paratera Tech Corp., Ltd for providing HPC resources that have contributed to the research results reported within this paper.

\* fengx463@alumni.umn.edu
† zymeng@hku.hk

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

SUPPLEMENTAL MATERIAL

**Scalable Hybrid quantum Monte Carlo simulation of QED$_3$ coupled to fermions on GPU**

In this supplemental material, we provide the technical details concerning the QMC measurements of conserved currents (Sec. SII) and the finite size scaling of the order parameters at the easy-plane DQCP (Sec. SIII), and the derivation of the fermionic correlation functions (Sec. SIV).

## SI. Calculation of Fermionic Force for QED$_3$

The fermionic force in the Hamiltonian dynamics can be derived in the following way,

$$F_{i,\tau} = \frac{\partial}{\partial \phi_{i,\tau}} \eta^\dagger \left( M^\dagger(\phi) M(\phi) \right)^{-1} \eta \tag{S1}$$

$$= - \eta^\dagger \left( M^\dagger(\phi) M(\phi) \right)^{-1} \left[ M^\dagger(\phi) \frac{\partial}{\partial \phi_{i,\tau}} M(\phi) + \left( \frac{\partial}{\partial \phi_{i,\tau}} M^\dagger(\phi) \right) M(\phi) \right] \left( M^\dagger(\phi) M(\phi) \right)^{-1} \eta, \tag{S2}$$

where $i$ is spacial lattice site position. Here, the $\left[ M^\dagger M \right]^{-1} \eta$ can be computed by the pCG algorithm as introduced in the main text, with complexity of $O(N_\tau V_s)$. This can be expressed as a vector $X = \left[ X_1^\mathsf{T}, X_2^\mathsf{T}, \ldots, X_{N_\tau}^\mathsf{T} \right]^\mathsf{T} := \left[ M^\dagger M \right]^{-1} \eta$. Then, given the definition of the $M$ matrix in the main text, the fermionic force can be straightforwardly computed as

$$F_{i,\tau} = X^\dagger \left[ M^\dagger(\phi) \left( \frac{\partial}{\partial \phi_{i,\tau}} M(\phi) \right) + \left( \frac{\partial}{\partial \phi_{i,\tau}} M^\dagger(\phi) \right) M(\phi) \right] X \tag{S3}$$

$$= \sum_{\tau'=1}^{N_\tau-1} \left[ X_{\tau'}^\dagger \left( B_{\tau'}^\dagger \frac{\partial B_{\tau'}}{\partial \phi_{i,\tau}} \right)^\dagger X_{\tau'} + X_{\tau'}^\dagger \left( B_{\tau'}^\dagger \frac{\partial B_{\tau'}}{\partial \phi_{i,\tau}} \right) X_{\tau'} + X_{\tau'+1}^\dagger \left( -\frac{\partial B_{\tau'}}{\partial \phi_{i,\tau}} \right) X_{\tau'} + X_{\tau'}^\dagger \left( -\frac{\partial B_{\tau'}}{\partial \phi_{i,\tau}} \right)^\dagger X_{\tau'+1} \right]$$

$$+ X_{N_\tau}^\dagger \left( B_{N_\tau}^\dagger \frac{\partial B_{N_\tau}}{\partial \phi_{i,\tau}} \right)^\dagger X_{N_\tau} + X_{N_\tau}^\dagger \left( B_{N_\tau}^\dagger \frac{\partial B_{N_\tau}}{\partial \phi_{i,\tau}} \right) X_{N_\tau} + X_1^\dagger \left( \frac{\partial B_1}{\partial \phi_{i,N_\tau}} \right) X_{N_\tau} + X_{N_\tau}^\dagger \left( \frac{\partial B_1}{\partial \phi_{i,N_\tau}} \right)^\dagger X_1. \tag{S4}$$

The remaining step is to evaluate $\frac{\partial B_{\tau'}}{\partial \phi_{i,\tau}}$. For symmetric checkerboard decomposition of the form $B_\tau = \prod_n e^{V_{\tau,n}/2}$ as shown in the main text, its derivative with respect to $\phi_i$ is

$$\frac{\partial B_{\tau'}}{\partial \phi_{i,\tau}} = \delta_{\tau\tau'} \sum_m \prod_{l<m} e^{V_{\tau,l}/2} \frac{\partial e^{V_{\tau,m}/2}}{\partial \phi_{i,\tau}} \prod_{n>m} e^{V_{\tau,n}/2}. \tag{S5}$$

Thus, each term in Eq. (S4) can be expressed as a series of matrix-vector multiplications, where the matrix is of the form $e^{V_{\tau,n}}$. Each matrix-vector multiplication has a time complexity of $O(N_\tau V_s)$. So, the total time complexity of computing the fermionic force is $O(N_\tau V_s)$.

## SII. Benchmark data

As mentioned in the main text, the HQMC algorithms applied to the QED$_3$ has an ergodicity issue, in the sense that the negative det $M$ region cannot be reached going from positive det $M$ region if only HQMC updates are proposed. However, we find that if $J$ is small enough, the negative det $M$ region will have a negligible possibility of being sampled, thus the HQMC method is valid for such values of $J$. In this section, we use the DQMC algorithm, which is ergodic, to benchmark HQMC and find the region of $J$ where the HQMC result ae-

grees with the DQMC result. We will show that the $J = 1.25$ we chose throughout the paper is inside such valid region.

Fig. S1 shows the averaged sign of det $M$ as a function of $J$ computed by DQMC for the non-compact QED$_3$. It shows that, at large $J$ where the gauge field is highly dynamic (great fluctuation along $\tau$ direction), the negative det $M$ samples are important and lower the average sign to be $< 1.0$. But as $J$ decreases to below 2.0 where the gauge field becomes more static, the sign is almost 1.0. Thus, the negative det $M$ samples are negligible.

Next, we compare the HQMC result with the DQMC result

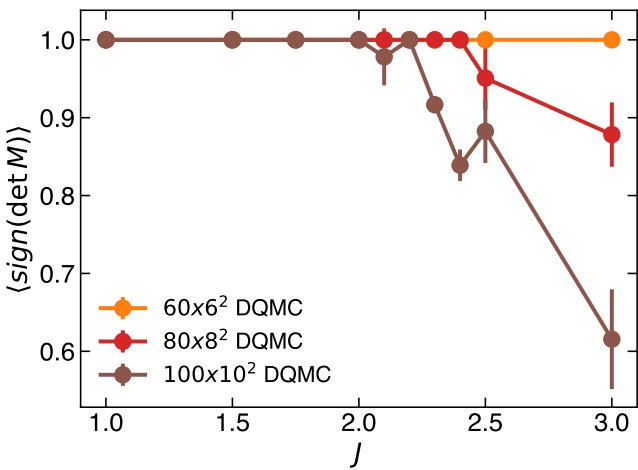

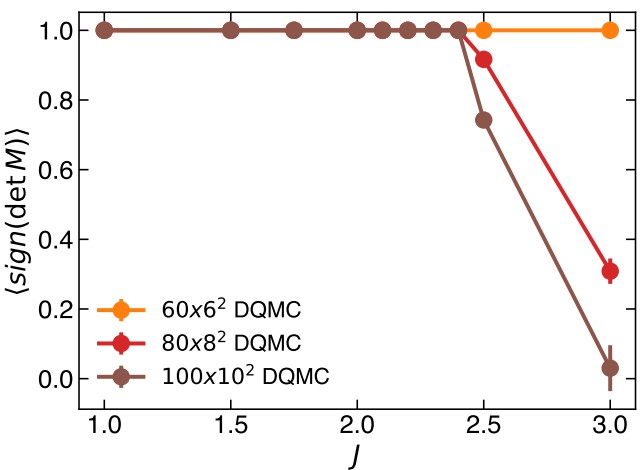

FIG. S1. The average sign of det $M$ as a function of $J$ for various lattice sizes shown as $N_\tau \times L^2$ computed for the non-compact QED$_3$ with $K = 0$, by DQMC method. For $J \lesssim 2.0$, the det $M$ are almost all positive. Thus, the ergodicity issue of HQMC is not severe in this regime of $J$.

FIG. S3. The average sign of det $M$ as a function of $J$ for various lattice sizes shown as $N_\tau \times L^2$ computed for the compact QED$_3$ with $K = 1$, by DQMC method. For $J \lesssim 2.2$, the det $M$ are almost all positive. Thus, the ergodicity issue of HQMC is not severe in this regime of $J$.

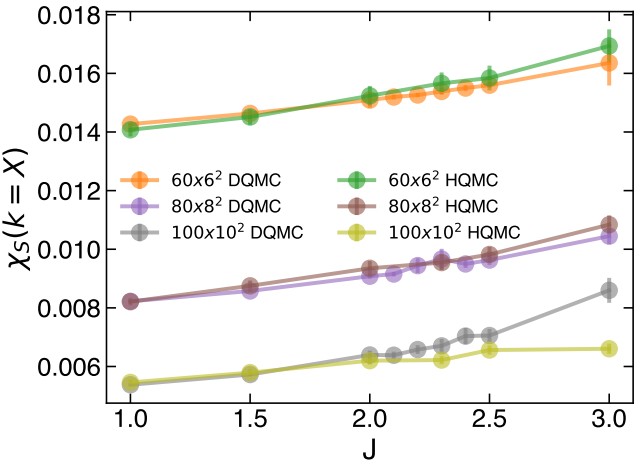

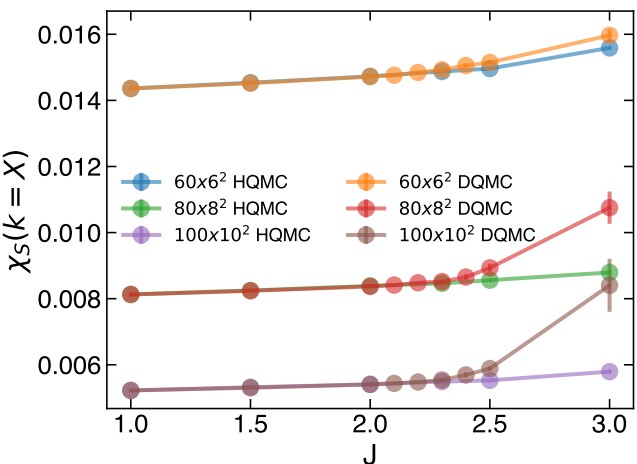

FIG. S2. The spin structure factor, which is Fourier transformation of spin-spin correlation function, evaluated at $X = (\pi, \pi)$ point, as a function of $J$, for various lattice sizes $N_\tau \times L^2$, computed for the non-compact QED$_3$ with $K = 0$. The plot compares the results between HQMC and DQMC. For $J \lesssim 2.0$, the spin structure factor from HQMC agrees well with that from DQMC; they only deviate from each other at large $J$, which is attributed to the ergodicity issue of the HQMC algorithm. This shows that in the compact QED$_3$ coupled with fermions, the ergodicity issue of HQMC is negligible for small $J$ deep in the putative $U(1)$ DSL phase.

FIG. S4. The spin structure factor, which is Fourier transformation of spin-spin correlation function, evaluated at $X = (\pi, \pi)$ point, as a function of $J$, for various lattice sizes $N_\tau \times L^2$, computed for the compact theory with $K = 1$. The plot compares the results between HQMC and DQMC. For $J \lesssim 2.2$, the spin structure factor from HQMC agrees well with that from DQMC; they only deviate from each other at large $J$, which is attributed to the ergodicity issue of the HQMC algorithm. This shows that in the compact QED$_3$ coupled with fermions, the ergodicity issue of HQMC is negligible for small $J$ deep in the putative $U(1)$ DSL phase.

for various $J$. Fig. S2 shows the spin structure factor, a Fourier transformation of the spin-spin correlation function, evaluated at $X = (\pi, \pi)$ point. This observable serves as a detect of the antiferromagnetic (AFM) order. In this figure, in the large $J$ region, HQMC result deviates from the DQMC result as expected since the negative det $M$ gauge configurations are important and not accessible to HQMC updates. But for $J \lesssim$

2.0, they agree with each other, indicating the ergodicity issue is negligible for these $J$.

Benchmarking HQMC with DQMC on compact QED$_3$ is similar to that on non-compact QED$_3$ which are shown in Fig. S3 and Fig. S4. Again, in the small $J$ region of around $J \lesssim 2.2$, the signs of the det $M$ are almost always positive, the AFM order parameters between HQMC and DQMC agree

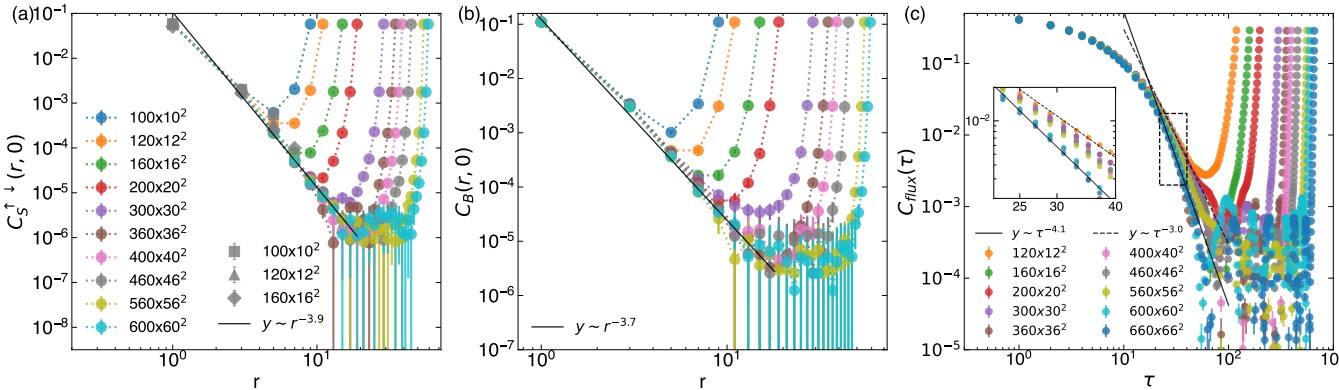

FIG. S5. The correlation functions for compact QED$_3$ with $K = 1$ and $J = 1.25$. **(a)** shows the log plot of the spin-spin correlation $C_S^{\uparrow\downarrow}(r, 0)$ as a function of distance $r$, for various cubic lattice sizes shown as $N_\tau \times L^2$ in the legend. To avoid the even-odd oscillation in the finite-size data, we only plot the data of the distance $r$ = odd points. The gray square, triangular and diamond dots show the results computed from the DQMC to benchmark the HQMC result of the corresponding lattice sizes. At the large distance and the large system sizes, the data are manually fitted by the black line, which displays a power law decay with scaling dimension of around 3.9. **(b)** shows the log plot of the bond-bond correlation (along $\hat{x}$ direction) as a function of distance $r$. The legend is the same as that in (a). At the large distance and the large system sizes, the manually fit line of the bond-bond correlation also displays a power law decay as $1/r^{3.7}$. **(c)** shows the log plot of the flux-flux correlation as a function of imaginary time $\tau$. At the large distance and the large system sizes, the black solid line shows the manual fitting which displays a power law decay as $\tau^{-4.1}$. The black dashed line shows the $\tau^{-3.0}$ power law behavior for comparison.

with each other, and thus the ergodicity issue of HQMC is negligible.

## SIII. Data for the compact QED$_3$ model

In this section, we present the results of the spin, bond and flux correlation for the compact QED$_3$ model, which is parallel to the results of noncompact QED$_3$ in the main text. We will again analyze the scaling behavior of these observables and demonstrate how they reveal the conformal field information in the $U(1)$ DSL critical phase.

Fig. S5(a) shows the log plot of the spin-spin correlation as a function of lattice distance $r$ for compact QED$_3$ at $J = 1.25$, for various lattice sizes. The noise level is at around $7 \times 10^{-6}$. As lattice size increases, the log plot of the spin-spin correlation function consistently displays a linear decay behavior at large distance $r$, until it reaches the noise level. This agrees with the power law decay behavior of the fermionic observable, which manifests the property of the $U(1)$ DSL critical phase. The linear decay is manually fitted by the black solid line which gives a power of around 3.9. This agrees with the scaling dimension of fermion bilinear operators $2\Delta_{\text{adj}} \in (3.5, 3.9)$ predicted by the CFT, where the fermion bilinear operators serve as the adjoint representation of the emergent $SU(4)$ symmetry, as introduced in the main text and Refs. [3, 17]. In Fig. S5(a), we also plot the DQMC results for several small lattice sizes to benchmark the HQMC results, and they agree well with each other.

Similarly, in Fig. S5(b) which shows the log plot of the bond-bond correlation as a function of distance $r$, the data points for various lattice sizes also consistently converge to a

straight line, until they reach a noise level of $2 \times 10^{-5}$. The points are large distance $r$ are manually fitted with the black solid line, which gives the power of around 3.7. This again agrees with the scaling dimension of fermion bilinear operators $2\Delta_{\text{adj}} \in (3.5, 3.9)$ predicted by the CFT.

Fig. S5(c) shows the log plot of the flux-flux correlation as a function of the imaginary time $\tau$ for compact QED$_3$ model with $K = 1$ and $J = 1.25$. As lattice size increases, the flux-flux correlation at large $\tau$ consistently converge to a clear linear behavior before they reach a noise level of around $2 \times 10^{-3}$. Such limit behavior is again manually fitted by the black solid line, which gives a power of around 4.1, and is distinct from the dashed line with the power of 3.0. This verifies the fact that the scaling dimension of the conserved current in $U(1)$ DSL phase is $\Delta_J = 2$.

In a nutshell, the results of spin, bond and magnetic flux correlation function of compact QED$_3$ does not have an essential difference from the results of noncompact QED$_3$ shown in the main text. This shows that, even though in theory the monopole operator in compact QED$_3$ may create relevant perturbation, gap out the gauge field and destroy the $U(1)$ DSL phase at any $J$, in actual numerical simulation, such signature is not seen at a finite lattice size of up to $N_\tau \times L^2 = 660 \times 66^2$ at $J = 1.25$.

## SIV. Derivation of fermionic correlation functions

In this section we give a detailed derivation of the fermionic correlation functions measured in HQMC, which includes the spin correlation function and bond correlation function.

As introduced in the main text, the spin operator for $N_f = 2$

is defined as

$$S^{\alpha\beta}(i) = \psi_{i\alpha}^{\dagger}\psi_{i\beta} - \frac{1}{2}\delta_{\alpha\beta}\sum_{\gamma}\psi_{i\gamma}^{\dagger}\psi_{i\gamma}. \tag{S6}$$

The bond operator along the nearest-neighbor bond in $\hat{x}$ direction is defined as

$$B_i = \sum_{\alpha}(\psi_{i,\alpha}^{\dagger}e^{i\phi_{ij}}\psi_{i+\hat{x},\alpha} + \text{h.c.}), \tag{S7}$$

and $\alpha, \beta \in \{\uparrow, \downarrow\}$ are spin flavor indices. Then, the spin correlation function is

$$C_S^{\uparrow\downarrow}(i,j) = \langle S^{\uparrow\downarrow}(i)S^{\downarrow\uparrow}(j)\rangle_{\psi\phi} \tag{S8}$$

$$= \langle \psi_{i\uparrow}^{\dagger}\psi_{i\downarrow}\psi_{j\downarrow}^{\dagger}\psi_{j\uparrow}\rangle_{\psi\phi} \tag{S9}$$

$$= \langle \bar{G}_{\uparrow}(i,j)G_{\downarrow}(i,j)\rangle_{\phi}, \tag{S10}$$

where the last row utilizes Wick's theorem, and the subscripts $\psi$ and $\phi$ mean the average over the fermionic field and the bosonic gauge field, respectively. The equal-time Green's functions are defined as

$$G_{\alpha}(i,j) = \langle \psi_{i\alpha}\psi_{j\alpha}^{\dagger}\rangle_{\psi}, \tag{S11}$$

$$\bar{G}_{\alpha}(i,j) = \langle \psi_{i\alpha}^{\dagger}\psi_{j\alpha}\rangle_{\psi} \tag{S12}$$

$$= \delta_{ij} - G_{\alpha}(j,i). \tag{S13}$$

In this problem, the Green's function of the two fermion flavors are equal $G_{\uparrow} = G_{\downarrow}$. So we will omit the flavor index in the Green's functions below. Similarly, the bond correlation function is

$$C_B(i,j) = \langle B_iB_j\rangle_{\psi\phi} - \langle B_i\rangle_{\psi\phi}\langle B_j\rangle_{\psi\phi}. \tag{S14}$$

Here,

$$\langle B_iB_j\rangle_{\psi\phi} = 2\langle\bar{G}(i,j+\hat{x})G(i+\hat{x},j)e^{i(\phi_{i,i+\hat{x}}+\phi_{j,j+\hat{x}})}$$
$$+\bar{G}(i,j)G(i+\hat{x},j+\hat{x})e^{i(\phi_{i,i+\hat{x}}-\phi_{j,j+\hat{x}})}$$
$$+\bar{G}(i+\hat{x},j+\hat{x})G(i,j)e^{i(-\phi_{i,i+\hat{x}}+\phi_{j,j+\hat{x}})}$$
$$+\bar{G}(i+\hat{x},j)G(i,j+\hat{x})e^{-i(\phi_{i,i+\hat{x}}+\phi_{j,j+\hat{x}})}\rangle_{\phi}$$
$$+4\langle\left[\bar{G}(i,i+\hat{x})e^{i\phi_{i,i+\hat{x}}} + \bar{G}(i+\hat{x},i)e^{-i\phi_{i,i+\hat{x}}}\right]$$
$$\left[\bar{G}(j,j+\hat{x})e^{i\phi_{j,j+\hat{x}}} + \bar{G}(j+\hat{x},j)e^{-i\phi_{j,j+\hat{x}}}\right]\rangle_{\phi}, \tag{S15}$$

where the coefficients come from the summation of the fermionic flavors.

$$\langle B_i\rangle_{\psi\phi}\langle B_j\rangle_{\psi\phi} = 4\langle\bar{G}(i,i+\hat{x})e^{i\phi_{i,i+\hat{x}}} + \bar{G}(i+\hat{x},i)e^{-i\phi_{i,i+\hat{x}}}\rangle_{\phi}$$
$$\langle\bar{G}(j,j+\hat{x})e^{i\phi_{j,j+\hat{x}}} + \bar{G}(j+\hat{x},j)e^{-i\phi_{j,j+\hat{x}}}\rangle_{\phi}, \tag{S16}$$

We denote the vector $b_i = \bar{G}(i,i+\hat{x})e^{i\phi_{i,i+\hat{x}}} + \bar{G}(i+\hat{x},i)e^{-i\phi_{i,i+\hat{x}}}$, then

$$C_B(i,j) = 2\langle\bar{G}(i,j+\hat{x})G(i+\hat{x},j)e^{i(\phi_{i,i+\hat{x}}+\phi_{j,j+\hat{x}})}$$
$$+\bar{G}(i,j)G(i+\hat{x},j+\hat{x})e^{i(\phi_{i,i+\hat{x}}-\phi_{j,j+\hat{x}})}$$
$$+\bar{G}(i+\hat{x},j+\hat{x})G(i,j)e^{i(-\phi_{i,i+\hat{x}}+\phi_{j,j+\hat{x}})}$$
$$+\bar{G}(i+\hat{x},j)G(i,j+\hat{x})e^{-i(\phi_{i,i+\hat{x}}+\phi_{j,j+\hat{x}})}\rangle_{\phi}$$
$$+4\left(\langle b_ib_j\rangle_{\phi} - \langle b_i\rangle_{\phi}\langle b_j\rangle_{\phi}\right), \tag{S17}$$

where the first term is the intrinsic fermionic correlation while the second term is the correlation from the gauge field fluctuation.