# Peer review of "Scalable hybrid quantum Monte Carlo simulation of U(1) gauge field coupled to fermions on GPU"

_SciPost Physics_

## Round 3 · Referee Report · Anonymous (Referee 1) · 2025-11-19

Strengths

(i) Addresses a long-standing problem whose importance is growing given the latest model and material studies.
(ii) Original and innovative methodological developments enabling record large simulations
(iii) Results substantially improve our understanding of the fate of U(1) gauge field couplied to fermions

Weaknesses

(i) The stability of the numerical results against certain parametric variations needs to be demonstrated more thoroughly.
(ii) Complementary and additional quantities need to be analyzed to conclusively establish the obtained results

Report

The manuscript “Scalable hybrid quantum Monte Carlo simulation of U(1) gauge field coupled to fermions on GPU” by Feng et al. is a formidable and timely attack on a pressing and long-standing problem concerning the fate of the critical theory of U(1) gauge field coupled to fermions. With the arrival of new models and candidate materials promising to host a U(1) Dirac spin liquid ground state, this precarious issue has been addressed with an innovate numerical development proposed in this paper. The results obtained on record-large system sizes for the scaling dimension of the relevant operators lends an increased confidence towards establishing it as a conformal field theory for the physically interesting case of the number of fermion flavors.

The manuscript is well-written and I would recommend its publication in SciPost Physics after the authors have answered the following concerns/questions and addressed these issues in an appropriately revised manuscript:

(i) The manuscript would benefit from quantitative measurements of autocorrelation times for gauge fields, fermionic bilinears, and energy. Could the authors provide τ_L or acceptance-rate scaling with L, and comment on whether critical slowing down is suppressed or only reduced?

(ii) The authors fix N_τ = 10 L (thus β = L).

Questions: Why is this particular scaling chosen?
Has the dependence on Δτ been checked?
Are the power-law exponents stable as Δτ → 0?
Could the authors provide a small-Δτ extrapolation or a consistency check?

(iii) While Fig. 2(b) shows small-L comparisons, no quantitative error analysis is given.
Could the authors provide explicit differences between HQMC and exact/ED or DQMC observables (e.g., energies, correlation functions) to demonstrate correctness and absence of systematic biases?

(iv) The scaling fits are done on a small number of large-distance points due to noise limitations.
Can the authors quantify the fitting range and systematic drift of exponents as the fitting window is varied? Are the statistical uncertainties (including covariance between data points) properly propagated?

(v) Since DSL/QED3 physics emerges only in the infrared, the accessible L may still be in a crossover regime. Are there indicators (e.g., scaling collapse, monotonic drift of exponents with 1/L) that the simulations have indeed reached the asymptotic scaling regime?

(vi) The paper interprets matching exponents for spin and bond operators as evidence for emergent SU(4). Can the authors provide a quantitative comparison (e.g., exponent difference Δ_spin − Δ_bond with uncertainty)? Is the agreement within error bars expected theoretically at these lattice sizes? Are there other operators (e.g., adjoint vs singlet channels) that could strengthen the SU(4) conclusion?

(vii) The τ⁻⁴ behaviour is associated with conserved current scaling Δ_J = 2. Can the authors distinguish τ⁻⁴ from possible τ⁻³ (free photon) or τ⁻⁵ behaviors within error bars? Are the results sensitive to gauge coupling J or to anisotropy in temporal vs spatial discretization?

Requested changes

Please see points raised in the Report

Recommendation

Publish (surpasses expectations and criteria for this Journal; among top 10%)

---

## Editorial Decision

in_refereeing